# Influence of PMMA on All-Inorganic Halide Perovskite CsPbBr_3_ Quantum Dots Combined with Polymer Matrix

**DOI:** 10.3390/ma12060985

**Published:** 2019-03-25

**Authors:** Lung-Chien Chen, Ching-Ho Tien, Zong-Liang Tseng, Yu-Shen Dong, Shengyi Yang

**Affiliations:** 1Department of Electro-Optical Engineering, National Taipei University of Technology, Taipei 10608, Taiwan; chtien@mail.ntut.edu.tw (C.-H.T.); t107658051@ntut.org.tw (Y.-S.D.); 2Department of Electronic Engineering, Ming Chi University of Technology, New Taipei City 24301, Taiwan; zltseng@mail.mcut.edu.tw; 3School of Physics, Beijing Institute of Technology, Beijing 100081, China

**Keywords:** perovskite, quantum dots, CsPbBr_3_, PMMA

## Abstract

The poor stability of CsPbX_3_ quantum dots (QDs-CsPbX_3_) under wet conditions is still considered to be a key issue. In order to overcome this problem, this study presents a high molecular weight polymer matrix (polymethylmethacrylate, PMMA) incorporated into the QDs-CsPbBr_3_ to improve its stability and maintain its excellent optical properties. In this study, the Cs_2_CO_3_, PbO, Tetrabutylammonium Bromide (TOAB) powder, oleic acid, and toluene solvent were uniformly mixed and purified to prepare high-quality QDs powders. Then, hexane was used as a dispersing agent for the QD powder to complete the perovskite QDs-CsPbBr_3_ solution. Finally, a solution with different proportions of quantum dots CsPbBr_3_ and PMMA was prepared and discussed. In the preparation of thin films, firstly, a thin film with the structure of glass/QD-CsPbBr_3_/PMMA was fabricated in a glove box using a well-developed QDs-CsPbBr_3_ solution by changing the ratio of CsPbBr_3_:PMMA. The material analysis of QDs-CsPbBr_3_ thin films was performed with photoluminescence (PL), transmittance, absorbance, and transmission electron microscopy (TEM). The structures and morphologies were further examined to study the effect of doped PMMA on perovskite QDs-CsPbBr_3_.

## 1. Introduction

Colloidal quantum dots (QDs) have received extensive attention owing to their high photoluminescence quantum yield (PLQY), tunable emission wavelength, broad band absorption, and narrow emission band [1,2,3,4,5,6,7]. Due to its unique quantum confinement effect [7,8,9], researchers can achieve bandgap tuning by adjusting the emission color of QDs size and content [10,11]. Given these excellent optoelectronic properties, QDs are widely used in solar cells [12,13], light-emitting diodes (LEDs) [14,15], photodetectors [16,17], and lasers [18,19]. QDs are also used in nuclear medicine and in medical imaging [20,21,22,23]. In recent years, the power conversion efficiency of lead-halide perovskite in solar cells has rapidly climbed from 3.81% to 23.7% [24,25], and it took only ten years to achieve this breakthrough. The great success of lead-halide perovskite in the photovoltaic industry has also prompted researchers to explore the application of lead-halide perovskite in other related fields. The light absorbing material perovskite has a common structural formula ABX_3_, and A may be an organic cation CH_3_NH_3_^+^(MA), HC(NH_2_)_2_^+^(FA), or an inorganic cation Cs^+^; B is usually a divalent cation Pb^2+^, Sn^2+^; and X is a halogen anion Cl^−^, Br^−^, I^−^. However, commonly used organic-inorganic hybrid perovskites contain the organic cation MA or FA, which are easily susceptible to degradation under oxygen and moist heat [26,27,28]. In order to solve the material stability problem, the all-inorganic cesium lead halide perovskite CsPbX_3_ (X = I, Br, Cl) has begun to attract attention, which is considered the most suitable substitute for organic-inorganic perovskites. The all-inorganic perovskite CsPbX_3_ (X = Cl, Br, I) is expected to replace the conventional fluorescent materials because it has a narrow half-width of 20 to 40 nm, a high PLQY of up to 90%, and a simple preparation method. By adjusting the ratio of halogen element X (X = Cl, Br, I), the perovskite QDs luminescence wavelength can be adjusted from 380 to 780 nm, which realizes an adjustable full visible light spectrum [29,30,31]. The preparation of perovskite QDs in LED devices can achieve a breakthrough of more than 110% in the National Television System Committee (NTSC) color gamut, and a better color rendering performance [32]. However, the stability of perovskite QDs is insufficient. It is generally believed that perovskite QDs are susceptible to the influence of water and oxygen molecules in the air, causing decomposition and destruction of their structures, decreasing the luminous efficiency, shortening the luminescence lifetime, and easily leading to failure, especially red light perovskite QDs (CsPbI_3_) [33]. The use of the high molecular polymer matrix for encapsulation isolates the perovskite QDs from water and oxygen, which is beneficial for the stability of the perovskite QDs, such as PMMA [1], poly(maleic anhydride-alt-1-octadecene) (PMA) [34], or inorganic protective layer (SiO_2_ [35,36,37,38], Al_2_O_3_ [39] and TiO_2_ [40], etc.). In this study, two kinds of perovskite QDs-CsPbX_3_ were prepared: one is the QDs solutions uniformly prepared from the QDs-CsPbX_3_ mixed with PMMA and the other one is the perovskite QDs-CsPbX_3_ films coated with PMMA as a protective layer. The properties of the two preparation methods were investigated. PMMA can protect the perovskite QDs and reduce the influence of moisture and oxygen in the atmosphere, thereby improving the stability of the perovskite QDs due to its high light transmittance and low excitation light.

## 2. Materials and Methods 

The perovskite CsPbBr_3_ solution was prepared by the chemical solution synthesis method. First, a Cs:Pb solution was prepared, and the Cs_2_CO_3_ powder (162.9 mg) (Echo Chemical Co., Ltd., Miaoli, Taiwan) and PbO powder (233.2 mg) (Echo Chemical Co., Ltd., Miaoli, Taiwan) were added to 5 mL of oleic acid (Echo Chemical Co., Ltd., Miaoli, Taiwan). The Cs:Pb solution was stirred until it was transparent at 160 °C using a hot plate stirrer. The Cs:Pb solution was placed in a circulator oven and heated to 120 °C for 30 min to bake the solution. Following this, 5 mL of toluene (Echo Chemical Co., Ltd., Miaoli, Taiwan) was injected for dilution. A total of 1 mL of the Cs:Pb solution was taken out, which was then poured into 15 mL of toluene and stirred at room temperature for 5 min. The Tetrabutylammonium Bromide (TOAB) powder (54.68 mg) (Echo Chemical Co., Ltd., Miaoli, Taiwan), 0.5 mL of oleic acid, and 2 mL of toluene were added together to blend, and stirred at room temperature ambient to obtain a Br source solution. The resulting Br source was injected into the Cs:Pb solution to obtain the unpurified perovskite CsPbBr_3_ solution. A total of 3 mL of ethyl acetate (Echo Chemical Co., Ltd., Miaoli, Taiwan) was added to the unpurified perovskite CsPbBr_3_ solution, followed by the centrifugal process with 6000 rpm for 20 min to separate the green precipitate from the unpurified perovskite CsPbBr_3_ precursor solution. The green precipitate was dried under vacuum for 12 h to remove the solvent to complete the purification step. The green precipitate CsPbBr_3_ powder was then dissolved in 50 μL of hexane, and vortexed for 5 min using an ultrasonic oscillating machine to prepare a CsPbBr_3_ perovskite QD solution. A mixture of 50 mg of PMMA powder (Uni-Onward Co., Ltd., New Taipei City, Taiwan) and 1 mL of toluene was stirred at 80 °C until the powder was completely dissolved. At this time, the solution was transparent and colorless, and it was left at room temperature for cooling to complete the PMMA solution. The QDs-CsPbBr_3_ solution and the PMMA solution were mixed in different proportions, which were QDs-CsPbBr_3_:PMMA = 1:2, 1:1, 2:1, and 3:1. Finally, the QDs-CsPbBr_3_:PMMA mixed solution was coated on a slide glass at 1000 rpm for 15 seconds to complete the glass/QDs-CsPbBr_3_:PMMA film preparation, as shown in Figure 1. In addition, the 50 μL of QD-CsPbBr_3_ solution was spin-coated on the glass substrate at 1000 rpm for 15 s to complete the preparation of the glass/QDs-CsPbBr_3_ film. Following this, 100 μL of the PMMA solution was spin-coated on the glass/QDs-CsPbBr_3_ sample at 1000 rpm for 15 s to complete the film preparation of the glass/QDs-CsPbBr_3_/PMMA, as shown in Figure 2.

The photoluminescence (PL) spectra was measured using a Protrustech UniRAM low temperature Raman/PL spectrophotometer system. The transmittance and absorbance spectra were measured using a Hitachi U-4100 UV/Vis/NIR Spectrophotometer (Hitachi, Tokyo, Japan). The size of perovskite QDs was characterized by transmission electron microscopy (TEM, Tecnai F30, Philips, Amsterdam, The Netherlands).

## 3. Results and Discussion

In order to investigate the color change of QDs-CsPbBr_3_-doped PMMA solution, different QDs luminescent color changes were produced at different QDs-CsPbBr_3_:PMMA solution ratios, which, from left to right, PMMA, QDs-CsPbBr_3_:PMMA = 1:2, QDs-CsPbBr_3_:PMMA = 1:1, QDs-CsPbBr_3_:PMMA = 2:1, QDs-CsPbBr_3_:PMMA = 3:1, CsPbBr_3_, are shown in Figure 3a. The excited light source is a CW semiconductor laser (Homemade, Taipei, Taiwan) with a 405 nm wavelength and 5 mW of light of output power. When laser excitation was not used, it could be observed that the color of the QDs-CsPbBr_3_ solution was semitransparent, and the PMMA solution was nearly transparent, indicating that both have high transparency characteristics. As the PMMA ratio was increased, the color of the solution of QDs-CsPbBr_3_ tended to be more transparent. When excited by a 405 nm laser, the color of the QDs-CsPbBr_3_ solution was observed to be green, and the PMMA solution was not luminescent. As the proportion of PMMA increased, the luminous intensity of QDs-CsPbBr_3_ weakened, but it still retained a vivid green light. Figure 3b shows the PL excitation spectra of perovskite QDs-CsPbBr_3_ solution, which was a mixture of CsPbBr_3_ and PMMA with four different doping ratios of QDs-CsPbBr_3_:PMMA solution. It can be observed that the photoexcitation wavelength barely changes after doping with PMMA. They are within the limits of error. The location of the PL peak of the QDs-CsPbBr_3_ solution was 513 nm. The QDs-CsPbBr_3_:PMMA with four different doping ratios of 1:2, 1:1, 2:1, and 3:1 were 512, 512, 512, and 513 nm, respectively. It can be observed that when the QDs-CsPbBr_3_ solution was doped with various ratios of PMMA solution, the location of the PL peak barely shifted. This means that the PMMA does not influence the band structure of the QDs-CsPbBr_3_, but surrounds and protects the QDs-CsPbBr_3_ to stabilize the material structure [41,42].

Figure 4a,b show the transmittance and absorbance spectra of PMMA, QD-CsPbBr_3_, and QDs-CsPbBr_3_:PMMA solutions with different ratios. The transmittances of all solutions demonstrated a window in the wavelength range of 350–520 nm due to the absorption of QD-CsPbBr_3_. Moreover, it can be observed that the QDs-CsPbBr_3_:PMMA solutions with 1:2, 1:1, and 2:1 ratios exhibit a good transmission–absorption ratio of about 80:20%, indicating that the incorporation of PMMA improves the contrast. On the other hand, the optical edges of the absorption spectra of PMMA solution, QDs-CsPbBr_3_ solution, and four QDs-CsPbBr_3_:PMMA solutions with different ratios of 1:2, 1:1, 2:1, and 3:1 are consistent. The location of the optical edge is 520 nm, corresponding to the band gap of the QDs-CsPbBr_3_. It was shown that when the QDs-CsPbBr_3_ concentration increases, the absorption rate increases. However, the band structures of all samples are not changed.

Figure 5a show the structural diagrams and photographs excited without and with a UV laser (405 nm, 5 mW) of the QDs-CsPbBr_3_ film, QDs-CsPbBr_3_/PMMA film, and QDs-CsPbBr_3_:PMMA film. It can be observed that the QD-CsPbBr_3_ film excited by a laser exhibits a bright pure-green light. The luminance of the QD-CsPbBr_3_/PMMA film and QD-CsPbBr_3_:PMMA film excited by the laser seem darker than that of the QD-CsPbBr_3_ film owing to the total reflection effect and Snell’s law. The refractive indexes of QD-CsPbBr_3_ and PMMA are 2.3 and 1.47, respectively [43,44]. Therefore, only around 12% of the light inside the CsPbBr_3_ film from the surface radiates to the outside, according to the light escape cone [45]. Figure 5b shows the PL spectrum of the QDs-CsPbBr_3_ film, QDs-CsPbBr_3_/PMMA film, and QD-CsPbBr_3_:PMMA film. The QD-CsPbBr_3_ film and QD-CsPbBr_3_/PMMA film peak locations were observed at 515 nm and 514 nm, respectively. Four different doping ratios of QD-CsPbBr_3_:PMMA films = 1:2, 1:1, 2:1, and 3:1 were 514, 515, 515, and 515 nm, respectively. Therefore, it can be observed that the PL peaks of the QD-CsPbBr_3_/PMMA film and QD-CsPbBr_3_:PMMA film almost do not shift when the PMMA doping ratio increases. This result was the same as the solution preparation characteristics.

Figure 6a,b show the transmittance and absorbance spectrums of perovskite QDs-CsPbBr_3_ film, which were the QDs-CsPbBr_3_ film, QDs-CsPbBr_3_/PMMA film, and QDs-CsPbBr_3_:PMMA films with four different doping ratios. It can be found that the transmittance of the QDs-CsPbBr_3_ film and QDs-CsPbBr_3_/PMMA film was about 80%–90% in the visible range. When the proportion of doped PMMA increases, the transmittance of the QDs-CsPbBr_3_:PMMA film also becomes higher. When the QDs-CsPbBr_3_:PMMA film = 1:2, the transmittance at a wavelength of 400–700 nm was higher than about 90%. However, when the QDs-CsPbBr_3_:PMMA film = 3:1, the transmittance rate drops below 80%, indicating that the QDs concentration ratio is too high, which affects the optical properties of the QDs-CsPbBr_3_:PMMA film. On the other hand, the absorption steps of QDs-CsPbBr_3_ film, QDs-CsPbBr_3_/PMMA film, and four different ratios of QDs-CsPbBr_3_:PMMA films prepared as thin films were observed at 520 nm, corresponding to the band gap of the QDs-CsPbBr_3_. This result was also the same as the solution samples.

In order to observe the lattice structure, actual dispersion, and particle size of QDs, TEM analysis was employed to compare the changes before and after doping with PMMA. The preparation of the samples was then applied as a droplet on a copper grid using a dropper, directly, and baked at 40 °C for 20 min for TEM analysis. The TEM images of perovskite QDs-CsPbBr_3_ and QDs-CsPbBr_3_:PMMA are shown in Figure 7. It can be seen from Figure 7a–d that the particle size of QDs-CsPbBr_3_ was about 15 nm, of which the average particle size was obviously larger, and the morphology and arrangement were less irregular. After doping with PMMA, the particle size of QDs-CsPbBr_3_:PMMA was about 5–10 nm, of which the average particle size was significantly smaller, and its morphology and arrangement were more regular, as shown in Figure 7e–h.

## 4. Conclusions

In conclusion, we successfully used Cs_2_CO_3_, PbO, TOAB powder and oleic acid, toluene, and other solvents to prepare high-quality QDs powder, and then used hexene as a dispersing agent for QDs powder to complete the perovskite QDs-CsPbBr_3_ solution. By doping PMMA, the optical transparency of QDs-CsPbBr_3_ can be increased. It can be seen from the TEM image that the particle size presented is smaller, and its morphology and arrangement are also more regular. It is particularly clear that the geometry of QD changes from rectangular to square. Due to the low photoluminescent background of PMMA, the excellent optical properties of QDs-CsPbBr_3_ can be maintained.

## Figures and Tables

**Figure 1 materials-12-00985-f001:**
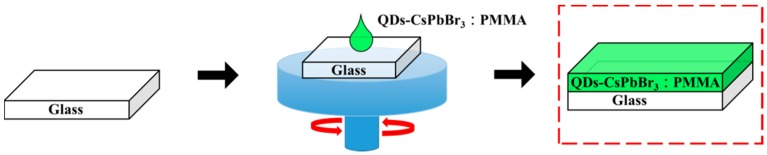
Flow chart of glass/QDs-CsPbBr_3_:PMMA film preparation.

**Figure 2 materials-12-00985-f002:**
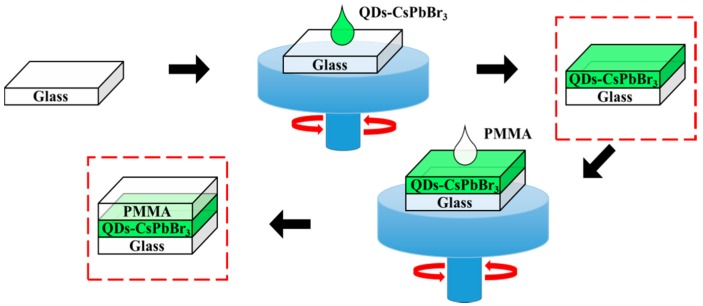
Flow chart of glass/QDs-CsPbBr_3_/PMMA film preparation.

**Figure 3 materials-12-00985-f003:**
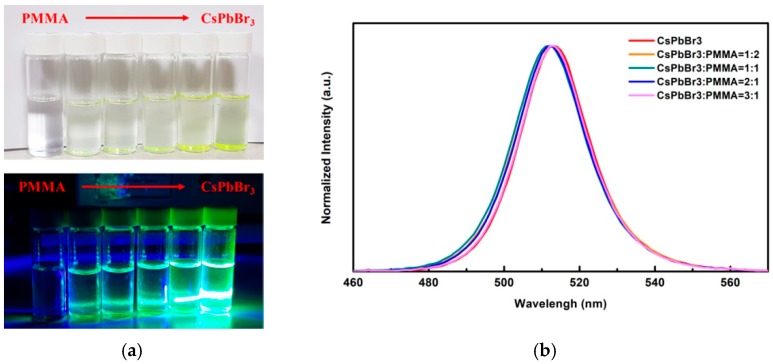
(**a**) Photographs of different ratios of QDs-CsPbBr_3_:PMMA solution with and without illumination, and (**b**) PL spectra of different ratios of QDs-CsPbBr_3_:PMMA solutions.

**Figure 4 materials-12-00985-f004:**
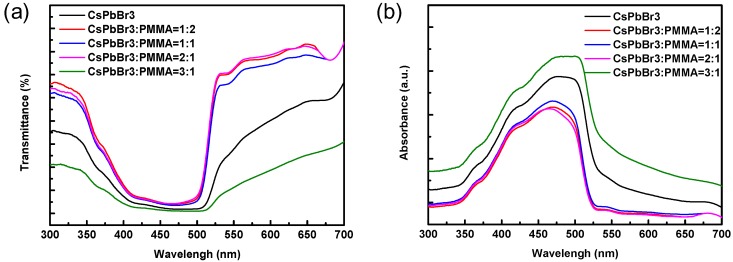
(**a**) Transmittance and (**b**) absorbance spectrums of the PMMA, QD-CsPbBr_3_, and four different ratios of QDs-CsPbBr_3_:PMMA solutions.

**Figure 5 materials-12-00985-f005:**
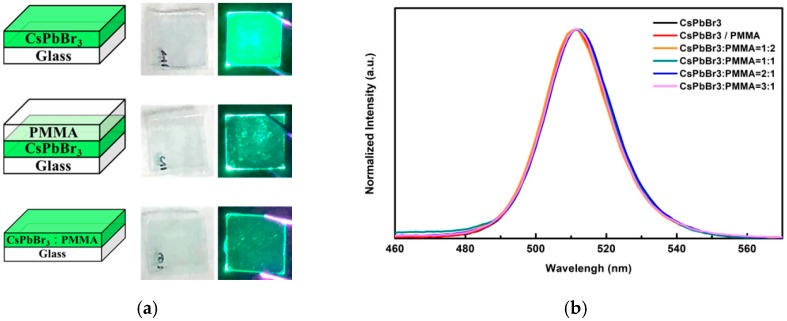
(**a**) Structural diagrams and photographs excited without and with a UV laser (405 nm, 5 mW) and (**b**) PL spectra of the QDs-CsPbBr_3_ film, QDs-CsPbBr_3_/PMMA film, and QDs-CsPbBr_3_:PMMA.

**Figure 6 materials-12-00985-f006:**
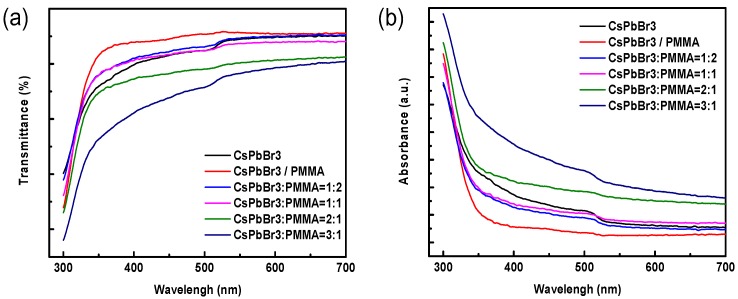
(**a**) Transmittance and (**b**) absorbance spectrums of the QD-CsPbBr_3_ film, QD-CsPbBr_3/_PMMA film, and four different ratios of QDs-CsPbBr_3_:PMMA films.

**Figure 7 materials-12-00985-f007:**
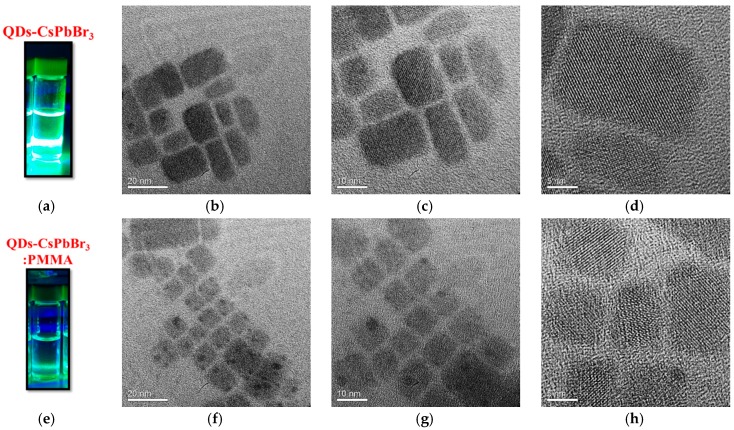
TEM images of (**a**–**d**) QDs-CsPbBr_3_ and (**e**–**h**) QDs-CsPbBr_3_:PMMA.

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
