# Peer review of "Influence of PMMA on All-Inorganic Halide Perovskite CsPbBr3 Quantum Dots Combined with Polymer Matrix"

_materials, 2019, doi:10.3390/ma12060985_

Round 1

Reviewer 1 Report

Review for the manuscript:

Entitled: " Influence of All-Inorganic Halide Perovskite CsPbBr3 Quantum Dots via Embedded in Polymer Matrix"

for Materials.

With ID: Materials-460647

Dear authors,

Thank you for your manuscript.

General comments

Comments for the Authors

This work is well within the scope of Materials and it may be of interest to most of the readers of this journal. It shows an introductory background material, sufficient for someone not an expert in this area to understand the context and significance of this work, with good, but few, references to follow. However, results section needs drastically improvement and justification. Furthermore, the English is not good, and the text needs proofread. Even the title of the manuscript has a syntax error (via Embedded in Polymer Matrix). For all the above and the specific comment below, I have opted to recommend a Major Revision for the current version of the article.

Specific comments

Title

Please revise the syntax of the manuscript’s title.

Abstract

P1, L23,24: ‘Observe the differences in photoluminescence intensity, absorbance, and material properties’ what is the meaning of this sentence? Please revise.

P1, L24: ‘For the thin film and material analysis of QDs-CsPbBr3, the analysis was..’ Two consecutive ‘analysis’, please revise this sentence.

P1, L26: ‘The properties of optical properties’ Please revise this sentence.

P1, L27: ‘further investigated to investigate’ Please revise this sentence.

Introduction

P1, L36: QD are also used in nuclear medicine and in medical imaging. Please expand the introduction section in order to include these disciplines with appropriate references.

P2, L45: ‘solve the material stable problem’ Maybe authors want to say ‘stability’?

P2, L48: ‘a simple synthesis method, and a cost saving,’ Please revise this sentence.

P2, L61-62: Please revise this sentence.

Materials and Methods

P2, L71: Why 120o C were selected?

P2, L72-73: ‘A 5mL’, ‘Next, A 1’ Please correct.

P2, L74: ‘TOAB’ Please cite “Tetrabutylammonium Bromide (TOAB)” the first time cited in the text, besides the abstract.

P2, L75: ‘which mixture’ Please correct.

P2, L76-77: “The completed Br source solution was injected into the Cs:Pb solution to complete the unpurified perovskite CsPbBr3 solution” Three “solution” in a row. Please revise.

P2, L85: “At this time, the solution was transparent and thick,” Please provide data for the final thickness of the produced films and the coating thickness (mg/cm2) of the QDs with the film. Are the obtained thicknesses appropriate for imaging applications and at which energies? The performance of a detector is determined by the luminescence efficiency and imaging resolution. Can authors provide efficiency and resolution results for their films?

P2, L88: ‘PMMA mixed solution was coated on the glass’ Why did the authors used glass as a substrate? What type of glass? It is fused silica (SiO2) substrates? Did authors measure the transmittance of the substrate? It is well known that the spatial resolution (for example in terms of the Modulation Transfer Function -MTF) is degraded due to the use of glass substrates (even thin ones).

Published results showed that there is a 40% degradation of the MTF on average, in the whole spatial frequency range, due to the glass substrate. The glass substrate has a low impact on resolution of the image at low frequencies (about 12% on average) and a high impact at medium and high frequencies (70% on average). Thus, for optimum performance composite flexible films should be prepared. For example, in a recent article (https://doi.org/10.1016/j.rinp.2017.05.011) PMMA/ZnCdSeS QD flexible films were prepared.

P3, L102-105: There is no reference regarding the excitation procedure and the time of measurement. After how many hours of drying on glass?

P3, L106-108: What type of transmittance measurements can support this finding??? It is well known that the PMMA solution is not luminescent under UV excitation. The luminous intensity was weakened because as the PPMA toluene mixture dries off and creates opacity in the PMMA. That’s PMMA should not dissolved in toluene but in MMA.

P3, L115: “short” -> “shorter” Of course this is not true since the excitation wavelength is not changing with the amount of PMMA.

P3, L115: ‘were 431, 432, 433, 433, 433, and 434 nm’ The excitation wavelength is not changing with the amount of PMMA. These data are within the limits of error.

P4, L120: ‘spectrums’ -> ‘spectra’.

P4, L121: ‘The transmittances of PMMA solution was’ Please revise.

P4, L120-130: The paragraph explaining Fig4a,b lacks scientific justification.

P4, L133: ‘an excitation photographs’ Please revise here and in Fig.5a caption. 

P4, L135: ‘pure green light but cannot penetrate the glass’ Please explain. Can authors discuss upon the refractive indexes of the two materials and the optical phenomena on their interface?

P4, L136: ‘QD-CsPbBr3/PMMA film and QD-CsPbBr3:PMMA film can penetrate and show better optical properties’ Isn’t it obvious?

P5, Fig5: Authors should also provide absolute irradiance optical spectra of the films, in units of μW/cm2/nm, by using various excitation intensities, in order to show the dependence of luminescence upon excitation intensity.

P5, L154: ‘indicating that the QDs concentration ratio is too high, which affects the optical properties’ Isn’t it obvious that an increase in the QD concentration will reduce optical transmittance? Furthermore, from figure 5a it is obvious that the transmittance of QD/PMMA is reduced (2nd image, middle) due to the opacity of PMMA/Toluene. The third image is expected to have even higher transmittance since QD is embedded within PMMA.

P5, L162: What was the sample preparation procedure for TEM analysis? On what substrate the samples were positioned?

P5, L164: ‘the particle size of’ Please provide a figure showing the QD particle size distribution along with the quartile deviation.

P6, L165: A challenge in the QDs film production processes is the aggregation of the nanoparticles in the polymer matrix, which can reduce fluorescence. Can authors discuss upon this issue?  

Conclusions

P6, L173-174: Is this conclusion statement really true?

Author Response

Reviewer 1 

This work is well within the scope of Materials and it may be of interest to most of the readers of this journal. It shows an introductory background material, sufficient for someone not an expert in this area to understand the context and significance of this work, with good, but few, references to follow. However, results section needs drastically improvement and justification. Furthermore, the English is not good, and the text needs proofread. Even the title of the manuscript has a syntax error (via Embedded in Polymer Matrix). For all the above and the specific comment below, I have opted to recommend a Major Revision for the current version of the article. 

Response: Thank you very much for review work.

Specific comments

Title

Please revise the syntax of the manuscript’s title.

Response: Thank you for your nice suggestion. The title has revised:

Influence of All-Inorganic Halide Perovskite CsPbBr3 Quantum Dots combined with Polymer Matrix

Abstract

P1, L23,24: ‘Observe the differences in photoluminescence intensity, absorbance, and material properties’ what is the meaning of this sentence? Please revise.

Response: Thank you for your valuable comments. The sentence has revised.

P1, L24: ‘For the thin film and material analysis of QDs-CsPbBr3, the analysis was..’ Two consecutive ‘analysis’, please revise this sentence.

Response: Thank you for your valuable comments. The sentence has revised.

P1, L26: ‘The properties of optical properties’ Please revise this sentence.

 Response: Thank you for your valuable comments. The sentence has revised.

P1, L27: ‘further investigated to investigate’ Please revise this sentence.

 Response: Thank you for your nice suggestion. The manuscript has revised (on page 2, line 23-27).

Introduction

P1, L36: QD are also used in nuclear medicine and in medical imaging. Please expand the introduction section in order to include these disciplines with appropriate references.

 Response: Thank you for your valuable comments. The manuscript has revised.

P2, L45: ‘solve the material stable problem’ Maybe authors want to say ‘stability’? 

Response: Revised.

P2, L48: ‘a simple synthesis method, and a cost saving,’ Please revise this sentence.

Response: Revised.

P2, L61-62: Please revise this sentence.

Response: Revised.

Materials and Methods

P2, L71: Why 120o C were selected? 

 Response: Just baking the solution to form the powder. The temperature should be lower 160 oC.

P2, L72-73: ‘A 5mL’, ‘Next, A 1’ Please correct. 

 Response: Revised.

P2, L74: ‘TOAB’ Please cite “Tetrabutylammonium Bromide (TOAB)” the first time cited in the text, besides the abstract.

 Response: Revised.

P2, L75: ‘which mixture’ Please correct. 

 Response: Revised.

P2, L76-77: “The completed Br source solution was injected into the Cs:Pb solution to complete the unpurified perovskite CsPbBr3 solution” Three “solution” in a row. Please revise.

 Response: Revised.

P2, L85: “At this time, the solution was transparent and thick,” Please provide data for the final thickness of the produced films and the coating thickness (mg/cm2) of the QDs with the film. Are the obtained thicknesses appropriate for imaging applications and at which energies? The performance of a detector is determined by the luminescence efficiency and imaging resolution. Can authors provide efficiency and resolution results for their films?

 Response: It is not exact mention. We have no measure the density. It means “colorless and transparent”.

P2, L88: ‘PMMA mixed solution was coated on the glass’ Why did the authors used glass as a substrate? What type of glass? It is fused silica (SiO2) substrates? Did authors measure the transmittance of the substrate? It is well known that the spatial resolution (for example in terms of the Modulation Transfer Function -MTF) is degraded due to the use of glass substrates (even thin ones).

Published results showed that there is a 40% degradation of the MTF on average, in the whole spatial frequency range, due to the glass substrate. The glass substrate has a low impact on resolution of the image at low frequencies (about 12% on average) and a high impact at medium and highfrequencies (70% on average). Thus, for optimum performance composite flexible films should be prepared. For example, in a recent article (https://doi.org/10.1016/j.rinp.2017.05.011) PMMA/ZnCdSeS QD flexible films were prepared.

Response: It is a slide glass. We are sorry that the Modulation Transfer Function (MTF) is not our major and topic. Please kindly understand we can not more statement about that.

P3, L102-105: There is no reference regarding the excitation procedure and the time of measurement. After how many hours of drying on glass?

Response: They are solution. The excited light source is a CW semiconductor laser with 405 nm of wavelength and 5 mW of light of output power.

P3, L106-108: What type of transmittance measurements can support this finding??? It is well known that the PMMA solution is not luminescent under UV excitation. The luminous intensity was weakened because as the PPMA toluene mixture dries off and creates opacity in the PMMA. That’s PMMA should not dissolved in toluene but in MMA.

Response: Thank you for your nice comment. We rewritten the sentence. (On page 3, Line 114-117)

P3, L115: “short” -> “shorter” Of course this is not true since the excitation wavelength is not changing with the amount of PMMA.

 Response: Revised.

P3, L115: ‘were 431, 432, 433, 433, 433, and 434 nm’ The excitation wavelength is not changing with the amount of PMMA. These data are within the limits of error.

 Response: Yes. Thank you for your valuable suggestion. The description in the manuscript has revised.

P4, L120: ‘spectrums’ -> ‘spectra’.

 Response: Revised.

P4, L121: ‘The transmittances of PMMA solution was’ Please revise.

 Response: Revised.

P4, L120-130: The paragraph explaining Fig4a,b lacks scientific justification.

Response: Thank you for your nice comment. We rewritten the paragraph. (On page 4, Line 130-140)

P4, L133: ‘an excitation photographs’ Please revise here and in Fig.5a caption. 

 Response: Revised.

P4, L135: ‘pure green light but cannot penetrate the glass’ Please explain. Can authors discuss upon the refractive indexes of the two materials and the optical phenomena on their interface?

Response: It is not exact mention. Thank you for your nice comment. The manuscript has rewritten. (On page 4, Line 137-142)

P4, L136: ‘QD-CsPbBr3/PMMA film and QD-CsPbBr3:PMMA film can penetrate and show better optical properties’ Isn’t it obvious?

 Response: It is not exact mention. Thank you for your nice comment. The manuscript has rewritten. (On page 4, Line 148-153)

P5, Fig5: Authors should also provide absolute irradiance optical spectra of the films, in units of μW/cm2/nm, by using various excitation intensities, in order to show the dependence of luminescence upon excitation intensity.

Response: The laser source is a semiconductor CW semiconductor laser with 405 nm of wavelength and 5 mW of light of output power.

P5, L154: ‘indicating that the QDs concentration ratio is too high, which affects the optical properties’ Isn’t it obvious that an increase in the QD concentration will reduce optical transmittance? Furthermore, from figure 5a it is obvious that the transmittance of QD/PMMA is reduced (2nd image, middle) due to the opacity of PMMA/Toluene. The third image is expected to have even higher transmittance since QD is embedded within PMMA.

 Response: It is not exact mention. Thank you for your nice comment. The manuscript has rewritten. (On page 4, Line 148-153)

P5, L162: What was the sample preparation procedure for TEM analysis? On what substrate the samples were positioned?

Response: The preparation procedure of samples is that dropped a droplet on a copper grid using a dropper, directly, and baked at 40 oC for 20 min for TEM analysis.

P5, L164: ‘the particle size of’ Please provide a figure showing the QD particle size distribution along with the quartile deviation.

 Response: Thank you for your nice comment. Unfortunately, we can not book the equipment to measure the particle size in this month. Therefore, please kindly understand we have no data in this moment.

P6, L165: A challenge in the QDs film production processes is the aggregation of the nanoparticles in the polymer matrix, which can reduce fluorescence. Can authors discuss upon this issue?  

 Response: Thank you for your suggestion. However, it is not the topic of this work. It may be our future work.

Conclusions

P6, L173-174: Is this conclusion statement really true?

 Response: Yes. We reworked all samples and measured by a new spectroscope. The phenomenon is same.

Reviewer 2 Report

All inorganic perovskite is the current hot issue. So, this work will add good contribution to this research field. Well written and presented. 

There are few typo mistakes:

line 42: "inorganic cation Cs-" should be changed to "inorganic cation Cs+"

line 45: "material stable problem" should be changed to "material stability problem"

line 122: "QDs-CsPbBr3 solution was less than 90%" is wrong it would be 80% according to the figure 4(a).

line 210: reference 9; need to add the title.

Author Response

Reviewer 2 

All inorganic perovskite is the current hot issue. So, this work will add good contribution to this research field. Well written and presented. 

There are few typo mistakes:

line 42: "inorganic cation Cs-" should be changed to "inorganic cation Cs+"

line 45: "material stable problem" should be changed to "material stability problem"

line 122: "QDs-CsPbBr3 solution was less than 90%" is wrong it would be 80% according to the figure 4(a).

line 210: reference 9; need to add the title.

Response: Thank you for your valuable suggestion and comments. The typo mistakes have revised.

Reviewer 3 Report

The authors present an investigation of the evolution of the optical properties of CsPbBr3 doped with different amount of PMMA, both in solution and thin films.

The study is physically sounded although it's not clear to me the origin of the PL peak shift, shown in both cases (solutions and films), depending in the amount of PMMA. 

I think that this point should be better explained in the article.

Author Response

The authors present an investigation of the evolution of the optical properties of CsPbBr3 doped with different amount of PMMA, both in solution and thin films.

The study is physically sounded although it's not clear to me the origin of the PL peak shift, shown in both cases (solutions and films), depending in the amount of PMMA. 

I think that this point should be better explained in the article.

Response: Yes. Thank you for your valuable suggestion. We review the manuscript again. All of PL peaks are very close. They are within the limits of error. It means that the PMMA does not influence the band structure of the QDs-CsPbBr3. It just surrounds and protects the QDs-CsPbBr3 to stabilize the material structure. Therefore, we rewritten the sentences. (on Page 3, Line 115-119)

Round 2

Reviewer 1 Report

Review for the manuscript:

Entitled: " Influence of All-Inorganic Halide Perovskite CsPbBr3 Quantum Dots via Embedded in Polymer Matrix"

for Materials.

With ID: Materials-460647_R1

Dear authors,

Thank you for your manuscript.

General comments

Comments for the Authors

My previous remarks were addressed, so I have opted to recommend the Acceptance of the manuscript.

Best regards